# Impact of Probiotics on the Performance of Endurance Athletes: A Systematic Review

**DOI:** 10.3390/ijerph182111576

**Published:** 2021-11-04

**Authors:** Jara Díaz-Jiménez, Eduardo Sánchez-Sánchez, Francisco Javier Ordoñez, Ignacio Rosety, Antonio Jesús Díaz, Manuel Rosety-Rodriguez, Miguel Ángel Rosety, Francisco Brenes

**Affiliations:** 1Campus Cádiz, Doctoral School of the University of Cádiz (EDUCA), Edificio Hospital Real (Primera Planta), Plaza Falla 8, 11003 Cádiz, Spain; 2Internal Medicine Department, Punta de Europa Hospital, 11207 Algeciras, Spain; eduardo.sanchez.sanchez.sspa@juntadeandalucia.es; 3Human Anatomy, School of Medicine, University of Cádiz, Plaza Fragela s/n, 11003 Cádiz, Spain; franciscojavier.ordonez@uca.es (F.J.O.); ignacio.rosety@uca.es (I.R.); 4Medicine Department, School of Nursing, University of Cádiz, Plaza Fragela s/n, 11003 Cádiz, Spain; antoniojesus.diaz@uca.es; 5Medicine Department, School of Medicine, University of Cádiz, Plaza Fragela s/n, 11003 Cádiz, Spain; manuel.rosetyrodriguez@gm.uca.es; 6Move-It Research Group, Biomedical Research and Innovation Insitute of Cádiz, Puerta del Mar University Hospital, University of Cádiz, Plaza Fragela s/n, 11003 Cádiz, Spain; miguelangel.rosety@gm.uca.es; 7Medicine Department, Puerta del Mar Hospital, 11009 Cádiz, Spain; salud.deporte@uca.es

**Keywords:** athletic, athletic performance, endurance training, probiotics, URTIs

## Abstract

Background: Probiotic supplements contain different strains of living microorganisms that promote the health of the host. These dietary supplements are increasingly being used by athletes to improve different aspects such as athletic performance, upper respiratory tract infections (URTIs), the immune system, oxidative stress, gastrointestinal (GI) problems, etc. This study aimed to identify the current evidence on the management of probiotics in endurance athletes and their relationship with sports performance. Methods: A systematic review of the last five years was carried out in PubMed, Scopus, Web of science, Sportdiscus and Embase databases. Results: Nine articles met the quality criteria. Of these, three reported direct benefits on sports performance. The remaining six articles found improvements in the reduction of oxidative stress, increased immune response and decreased incidence of URTIs. There is little scientific evidence on the direct relationship between the administration of probiotics in endurance athletes and sports performance. Conclusions: Benefits were found that probiotics could indirectly influence sports performance by improving other parameters such as the immune system, response to URTIs and decreased oxidative stress, as well as the monitoring of scheduled workouts.

## 1. Introduction

The gut microbiota is a complex community of microorganisms that stably colonize the intestinal surface. It consists of 1014 resident microorganisms including bacteria, archaea, viruses and fungi. Mainly, the gut microbiota in healthy individuals is dominated by four groups: Actinobacteria, Firmicutes, Proteobacteria and Bacteroidetes [1,2,3]. Their functions are very diverse, ranging from fermenting, digesting and absorbing nutrients necessary to obtain energy and maintain the homeostasis of the organism, synthesis of vitamins and essential amino acids, modulation of the immune system and control of oxidative stress and inflammatory responses, to the maturation of the nervous system through the secretion of neuroactive molecules, etc. [4,5]. It is modulated by different environmental factors, being sensitive to physiological and homeostatic changes, which can lead to alterations of the microbiota or dysbiosis. These factors include diet, stress, physical activity and exercise [6,7].

The relationship between gut microbiota and exercise is twofold, i.e., the microbiota can be altered by exercise, especially endurance exercise, due to increased oxidative stress [8], intestinal permeability, electrolyte imbalance, glycogen depletion, etc. [9]. In addition, the microbiota influences the individual’s ability to perform optimally during exercise due to its ability to take up energy, modulate the immune system, regulate GI health [10] and reverse the inflammatory response after exercise [11,12].

In recent years, there has been an increase in attendance at endurance sporting events such as marathons, ultramarathons, triathlons, cycling events, etc. This has led to an increase in the competitiveness of these events, as well as an increase in the active search for ergogenic aids by athletes with the aim of improving their sporting performance.

Some elite athletes take a diet low in plant fiber to slow gastric emptying and reduce intestinal problems during exercise, but this recommendation can lead to a decrease in the diversity and functionality of the intestinal microbiota [5,13]. Therefore, alternatives such as the use of probiotics have been studied.

Probiotics are living microorganisms that have health benefits when consumed in adequate amounts [14]. Recently, the use of these microorganisms in sport, to reduce the inflammatory response and episodes of gastroenteritis and URTIs among athletes, has increased [1,15]. This health improvement in athletes is expected to positively influence their physical performance [8].

One of the strongest lines of research in this field, among elite athletes, is the verification of how physical exercise, in general, modifies the microbiota. According to data reported by Scheiman, Veillonella causes an increase of 13% in endurance performance, due to a metabolic advantage by the colonization of lactate metabolizing organisms, transforming it into propionate. Therefore, this finding is very promising, currently investigating the creation of probiotic capsules composed of Veillonella to increase the population of this bacterium in the intestinal microbiota of athletes and thus improve their performance. Research is also being carried out in which fecal transplants from athletes with an abundance of this bacterium to other athletes is carried out to see if they promote the proliferation of these bacteria, and thus increase athletic performance [16].

In addition, it has been proven that the combination of probiotics and a diet rich in fiber reduces recovery days after intense periods of training, contributing to good performance [17] and avoiding the immunological effects of URTIs, and improving GI problems, psychological problems, oxidative stress, etc. of overtraining [18] and with it, sports performance.

The recommendation to use probiotics in these athletes must be backed by evidence, as no systematic review or meta-analysis has been found that assesses the quality or scientific rigor of the studies carried out to date.

Therefore, the aim of this review was to identify, following a systematic methodology, the current evidence on the management of probiotics in endurance athletes and their relationship with sports performance.

## 2. Materials and Methods

A systematic review of the literature was conducted. The results were obtained by direct online access through the following databases: PubMed, Web of Science (WOS), Scopus, Sportdiscus and Embase. The aim of this review was to address the following question: Does probiotic supplementation improve sports performance in endurance athletes?

To define the research question, the PICOS criteria (Table 1) were used.

We studied articles published in any country, by any institution or individual researcher, written in Spanish and English. These articles had to be accessible in full text through Open Access. The search was limited to articles published in the last five years (2016–2020).

The associated MeSH descriptors “endurance training”, “athletic performance”, “probiotics” and “prebiotics” were used for document retrieval. No subject classifiers (Subheadings) or Entry Terms were used. The search strategies used were as follows: “endurance training” AND “probiotics”; “athletic performance” AND “probiotics”; (“endurance training” OR “athletic performance”) AND “probiotics”; (“endurance training” OR “athletic performance”) AND “probiotics” NOT prebiotics.

The final selection of articles was made according to the following inclusion criteria: (a) studies published in journals indexed in international databases that were subjected to peer review, (b) access to the full text, (c) in humans and (d) written in English and Spanish; and exclusion criteria: (a) studies not based on the target population, (b) expert reports, letters from the editor, books, monographs, clinical narratives or systematic reviews or meta-analyses.

Due to the large number of articles found in the first search and as an assessment of quality, two sieving processes were carried out. The first one was based on the title and abstract, eliminating studies that were not on the topic of interest and whose populations were not endurance athletes. For the second screening, we used the quality questionnaire validated by Castro-Piñero and colleagues in 2009 [19].

## 3. Results

A total of 26 published articles were located, 8 (30.77%) PubMed, 1 (3.85%) Scopus, 2 (7.69%) Sportdiscus, 7 (26.92%) Web of Science and 8 (30.77%) Embase. Of these retrieved papers, 11 of them were redundant (42.31%).

Once the first screening was applied based on the title and abstract and compliance with the inclusion and exclusion criteria, 10 articles were selected. After assessing the quality of these papers using the quality questionnaire validated by Ruiz et al. in 2009, the 9 articles written in English were retained (Figure 1).

The results obtained showed different study parameters in the approach to the proposed topic (Table 2).

All the selected articles were double-blind randomized clinical trials (RCTs). The sample size in 8 of the 9 articles was below 50 subjects, and only one had 243 participants. Of the total number of publications, only four took both sexes as the sample. The remaining 5 only selected men.

Regarding the probiotics selected, only 2 used Lactobacillus plantarum P128 [21,23]. The rest were different strains such as Bifidobacterium animalis subsp. Lactis (10 × 10⁹) and Lactobacillus-Acidophilus (10 × 10⁹) [20], Bifidobacterium bifidum (W23), Bifidobacterium lactis (W51), Enterococcus faecium (W54), Lactobacillus acidophilus (W22), Lactobacillus brevis (W63) and Lactococcus lactis (W58) [25], Bifidobacterium longum subsp. longum Olympic No. 1 (OLP-01) [22], Bifidobacterium (B.) breve BR03 Streptococcus (S.) thermophilus FP4 at 5 bn live cell count (CFU) [26] and Lactobacillus casei Shirota (LcS) [28]. Two of the total articles used as intervention a combination of probiotics (Lactobacillus acidophilus, bifidobacterium bifidum and bifidobacteriumanimalis subsp. lactis) and the same probiotic + glutamine [24] and probiotic Bacillus coagulans GBI-30, 6086 + casein (protein) [27].

The probiotics were dispensed in different preparations with different amounts of colony-forming units (CFUs): capsules, bottles and sachets. In addition, they were used for different periods of time (varying between 12–140 days), at different times of the day (before breakfast [25], after breakfast [27], at each meal [22], breakfast and dinner [28], after exercise and before bedtime [23], before sleeping [20]) and had different administration (1–3 times a day).

The objectives addressed in the different articles are diverse. Each one investigated one or several aspects related to endurance athletes, such as oxidative stress, inflammatory response, sports performance, immune system and URTIs.

Five of the selected articles reported positive results regarding the administration of probiotics (determined), decreased oxidative stress (6–13% decrease in proinflammatory cytokines and 55% increase in anti-inflammatory cytokines) [23] and reduced prevalence of URTIs [28], due to reduced tryptophan degradation [25]. In particular, Epstein–Barr virus (EBV) and cytomegalovirus (CMV) citrate antibodies were reduced [28]. Probiotic and protein intake decreased recovery times [27].

Three of the nine articles discussed the direct influence of probiotics on athletic performance, indicating that it increases by modulation of the microbiota and metabolites, causing greater diversity in the microbiota [21], increased beneficial bacteria and decreased pathogens [22] and that it influences the mitigation of reduced performance and muscle tension days after exercise [26].

Of these five, two confirmed that the benefits found did not provide improvements in sports performance, but helped in a secondary way, as an increased training load (h/week) was observed. This was due to a reduction in UTIs, resulting in fewer interruptions to their season [25] and maintenance of total CD8 T-cell numbers, effector memory population and modulation of lymphocyte response [20], indicating an influence of probiotics on the immune system, reducing the risk of infection and training interruption.

Similarly, one of the publications indicates that the administration of the selected probiotics did not produce positive results, and that the concentration of eHsp72 (high levels of systemic stress) did not vary with respect to the proposed targets [24].

## 4. Discussion

Following the systematic review, the number of publications on the use of probiotics in endurance athletes is small and has limitations or possible biases. Only three of the nine studies found reported a direct relationship between probiotic consumption as an ergogenic aid and improved performance in endurance athletes, improving training management and adaptations to exercise, due to decreased muscle damage after sports activity [21,23,27]. Six of the articles found refer to an improvement in the immune system by reducing URTI episodes [25]. This improvement is due to a decrease in pro-inflammatory cytokines and an increase in anti-inflammatory cytokines, together with a reduction in oxidative stress [23]. This results in greater continuity in training planning [25,26,27] and the reduction of negative complications (inflammatory response [2], presence of URTIs [25]) derived from physical exercise, which helps to improve sports performance in these athletes (Figure 2).

### 4.1. Direct Link between Probiotic Consumption and Improved Performance

These findings are justified through several studies, such as the one conducted with endurance athletes supplemented with a probiotic yogurt (200 mL of yogurt diary including not less than 10 CFU/g Streptococcus thermophilus or *Lactobacillus* delbrueckiissp. bulgaricus) in which they had an increase in VO2max and aerobic power [29]. In addition, in adolescent female swimmers, the intervention with probiotics (400 mL of probiotic yogurt containing 4 × 10^10^ CFU/mL-*Lactobacillus* Acidophilus SPP, *Lactobacillus* Delbrueckii Bulgaricus, Bifidobacterium Bifidum and Streptococcus SalivarusThermnophilus), seems to have favored significant improvements in VO2max [30]. In the case of endurance runners, an improvement in fatigue times in a hot environment was obtained with probiotics containing specific strains (one capsule per day containing 45 billion CFU of *Lactobacillus*, Bifidobacterium and Streptococcus strains) [31].

The causal relationship between probiotic consumption and increased VO2max and/or fatigue has not been studied, but rather the changes produced in the supplemented vs. non-supplemented athlete. As a direct causal relationship has not been evaluated, this may be due to multiple factors, including fewer training interruptions due to less muscle damage, decreased URTIs, inflammatory cytokines, etc. More research is needed in this field to search for this causal relationship.

### 4.2. Indirect Link between Probiotic Consumption and Improved Performance

#### 4.2.1. Effect of Probiotics on the Immune System

In the study published by Vaisberg et al. in 2019, it can be observed how *Lactobacillus* casei Shirota (LcS), administered for a period of time equal to 30 days, showed benefits related to the modulation of the immune response, inflammatory and mucosal upper respiratory tract, presenting protective effects [32]. This result is similar to that found in another groups of athletes, where it can be seen how the administration of 400 mL of probiotic yogurt containing 4 × 10^10^ CFU/mL-*Lactobacillus* Acidophilus SPP, *Lactobacillus* Delbrueckii Bulgaricus, Bifidobacterium Bifidum and Streptococcus SalivarusThermnophilus reduced the number of episodes of URTIs in adolescent female swimmers [30]. In the study conducted by Haywood et al. in 2014, in elite rugby players, reduction in the duration and amount of URTI symptoms was obtained after administration of *Lactobacillus* helveticusLafti L 10, 2 × 10^10^ CFU [33].

#### 4.2.2. Effect of Probiotics on the Gastrointestinal System

GI problems are more prevalent in long distance and/or endurance runners (50–70%) in relation to other athletes, due to the redistribution of blood flow to the active muscles and skin, reducing irrigation at the GI level, which can cause abdominal pain during and after exercise. It can also damage the mucosa of the intestinal walls, increasing the probability of suffering permeability in the intestinal walls and blood loss, causing alterations in the GI protective microbiota, and generating endotoxemia [34]. The continuous movement up and down the intestine, the intake of fluids and nutrients and the malfunctioning of the GI system (diarrhea, vomiting, etc.) affects the assimilation of nutrients which can lead to decreased performance due to lack of energy, nutrients, etc. [35].

All these consequences have a negative influence on sports performance by increasing discomfort during training or competition. Therefore, several studies have been carried out with probiotics, looking for a solution to all these problems.

Among the articles found, active strains such as *Lactobacillus* acidophilus (CUL60 and CUL21), Bifidobacterium bifidum (CUL20) and Bifidobacterium animalis subsp. Lactis (CUL34) decrease the incidence and severity of GI symptoms [36].

There are few studies that report the positive influence of probiotics on the GI system in athletes. Supporting the previous article, we found others, such as a study with highly trained endurance athletes supplemented with L. salivarius (UCC118), which reported less intestinal hypermeability induced by exercise [37].

In another study, in which athletes studied were from various disciplines such as triathletes, cyclists and runners, supplementation with six different strains (Bifidobacterium bifidum W23, Bifidobacterium lactis W51, Enterococcus faecium W54, *Lactobacillus* acidophilus W22, *Lactobacillus* brevis W63 and Lactococcus lactis W58, minimum concentration was 2.5 × 10^9^ CFU/gr) administered in a probiotic reduced zonulin, allowing activation of the TLR2 signaling pathway, which enhances the intestinal barrier [38].

Due to the little evidence that is still available on the use of probiotics (*Lactobacillus* acidophilus, Bifidobacterium infantis, Bifidobacterium bifidum) to reduce GI problems in athletes, the EFSA (European Food Safety Authority) indicates that probiotics are not currently listed as an efficient and effective product in maintaining the proper functioning of the digestive tract and/or intestinal barrier [34].

#### 4.2.3. Effect of Probiotics on Oxidative Stress

Long distance and/or endurance sport can lead to an increase in oxidative stress, and with it an increase in inflammatory (IL-6) and anti-inflammatory (IL-10, IL-1ra, sTNFR) cytokines; therefore, it can be seen how probiotics are a widely used alternative to try to alleviate these effects, which results in an alteration of the immune system, influencing performance [39].

The intervention with probiotics containing *Lactobacillus* plantarum PS128 caused a decrease in oxidative stress in triathletes [23]. As there are articles that show that there is a sector of the population that has a genetic predisposition to present greater inflammatory profiles than the rest, which can affect the duration and severity of viral infections [40], this is an aspect that must also be taken into account, since genetic inheritance will also have an influence. Lactobacillus gasseri OLL2809 supplementation in university athletes prevented decreased NK cell action after strenuous exercise, resulting in improved defense against infection and elevated mood [41].

### 4.3. Adverse Events in the Use of Probiotics

After a review of the selected articles, no adverse events have been found following the use of probiotics in endurance athletes. Currently, there is no evidence on the occurrence of adverse events associated with probiotic consumption in the population, although the safety of probiotic administration in immunocompromised subjects is under investigation [42].

### 4.4. Other Benefits or Uses of Probiotics

Although the review shows the main benefits of probiotics in endurance athletes, others have not been investigated and may influence athletic performance. Mental health plays an important role in endurance athletes and mental illnesses such as depression and anxiety may influence performance. Although studies on the use of probiotics as an alternative treatment have been published, further research is needed because the existing evidence is of low to medium scientific quality [43,44], although the current results are very promising.

There are no studies on the effects of probiotic use in these athletes on the hormonal system, nervous system or intestinal metabolites.

### 4.5. Current Limitations in Evidence of Probiotic Use and Endurance Athletes

Among the limitations found in the different studies used for this review (see Table 1), it was observed that the samples of subjects are usually small (<50) [24,32,45], making it difficult to generalize the results. In addition, there is a gender bias, because most of the studies are conducted in men [32,45,46] or the number of women is very small [24,36], making it difficult to extrapolate the results to women. Sex steroid hormones play an important role in the immune system of the population. These hormones differ between the sexes. Testosterone decreases the activation of the immune system and estrogen enhances it, which may lead to differences in the immune response of athletes and alterations in their microbiome [47]. In addition, there may be differences in the composition of the microbiome (presence of different families of bacteria according to sex) [48].

The number of days of intervention is usually very small (≤30 days) [23,24,32,45], which can negatively influence the results. In addition, different strains are used (only two share the same strain with different CFU amounts [26,27]), forms of administration, amount administered [24,32], etc., making it impossible to compare data and have a broad evidence base.

There is no previous knowledge of the microbiome of these athletes, in order to know which strain or strains are those that may be altered, absent or can be improved by probiotic supplementation. It is necessary to know this microbiome in order to understand the changes produced with the administration of different strains of probiotics and the influence of sports practice.

Another of the biases found was the lack of the control of dietary intake. That is, there was no dietary pattern for both groups, placebo and intervention; therefore, the changes produced in their microbiota may be due to the diet they follow or the probiotics administered [24,32]. It should not be forgotten that there is strong evidence that nutrition, especially the consumption of prebiotics, and hydration cause changes in the gut microbiota.

## 5. Conclusions

There is little scientific evidence on the direct relationship between the administration of probiotics in endurance athletes and sports performance, so there is no evidence to recommend their use as a nutritional supplement.

Furthermore, the studies reviewed present a high variability in their methodology, especially the strains studied and the lack of prior analysis of the gut microbiota. Therefore, further research is needed in this field, with special attention paid to the changes that the gut microbiota may undergo in these athletes. The alterations present in the gut microbiota of these athletes may justify the use of one strain versus another.

Although a direct relationship has not been demonstrated, the use of probiotics has been related to a decrease in URTIs and an improvement in the immune system, which increases continuity in training and indirectly positively influences sports performance.

## Figures and Tables

**Figure 1 ijerph-18-11576-f001:**
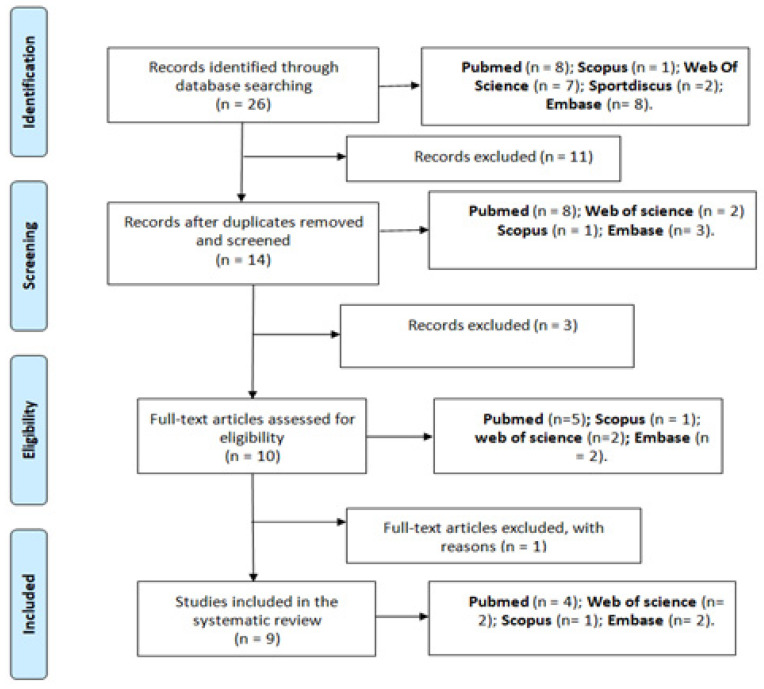
PRISMA (Preferred Reporting Items for Systematic Review and Meta-Analyses) diagram.

**Figure 2 ijerph-18-11576-f002:**
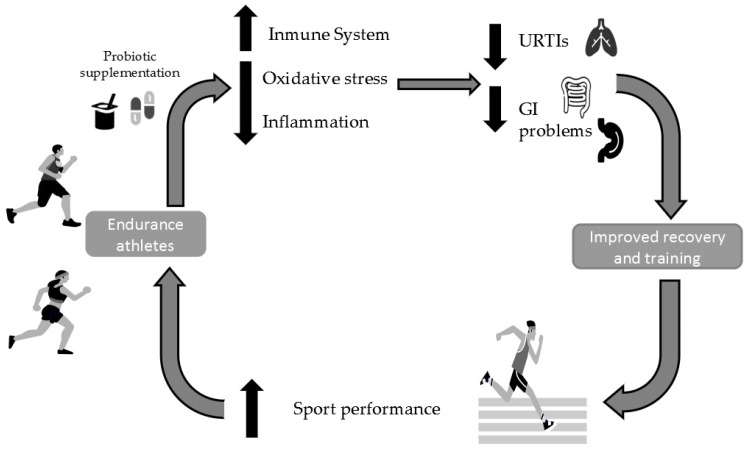
Mechanisms and outcomes of probiotic supplementation in endurance athletes. URTIs: upper respiratory tract infections; GI: gastrointestinal.

**Table 1 ijerph-18-11576-t001:** PICOS criteria (Population; Intervention; Comparison; Outcome; Study design).

P	Endurance Athletes
I	Probiotic supplementation
C	No supplementation
O	Improved sporting performance
S	Systematic review

**Table 2 ijerph-18-11576-t002:** Characteristics and main results of clinical trials on the use of probiotics in endurance athletes and improvement of sports performance.

Author, Year	Type of Study/Quality	Supplementation Procedure	Results	Conclusions
Batattinha et al., 2020 [20].	Double-blind, placebo-controlled RCT in 27 marathon runners (men) High quality = 5 N = 11–50	1 sachet of *Bifidobacterium-animalis-subs p.-Lactis *(10 × 10^9^ CFU)* and Lactobacillus-Acidophilus *(10 × 10^9^ CFU) + 5 g maltodextrin for 30 days before the race.	The total number of CD8 T lymphocytes was maintained in the probiotic group and the production of proinflammatory cytokines decreased, enhancing the immunomodulatory role of lymphocytes.	There were no differences between the two groups in relation to URTIs. The probiotic group modulates the lymphocyte response.
Huang et al. 2020 [21]	Double-blind, placebo-controlled RCT in 20 triathletes (male) High quality = 5 N = 11–50.	1 capful of *Lactobacillusplantarum PS128 *(1.5 × 10^10^*CFU*) for 4 weeks.	Increased endurance in the probiotic group. There were no significant differences in VO2max and body composition in the two groups.	LPS128 supplementation was associated with an improvement in endurance running performance through modulation of microbiota and related metabolites, but not in maximal oxygen uptake.
Lin et al. 2020 [22]	Double-blind, placebo-controlled RCT in 21 middle- and long-distance runners (7 women and 14 men) High quality = 5 N = 11–50	3 capsules/day for 5 weeks of *Bifidobacterium longum subsp. longum Olympic No. 1 (OLP-01)* (15 × 10^10^ CFU) after meals.	The OLP-01 group significantly increased the change in running distance of the 12-min Cooper test, with an increase in beneficial bacteria and decrease in pathogenic bacteria in the gut microbiota.	OLP-01 can be used as a sports nutrition supplement to enhance exercise performance.
Huang et al. 2019 [23]	Double-blind, placebo-controlled RCT in 34 triathletes. It was divided into study I (18 triathletes) and study II (16 triathletes). High quality = 5 N = 11–50	Study I, programmed training (triathlon preparation) and supplementation with *Lactobacillus plantarum PS128* lasting 4 weeks. Study II, specialized training and supplementation with *Lactobcillusplantarum PS128* lasting 3 weeks and dietary recommendations (30–40 g of carbohydrates and 500–1000 mL of water/hour) during the triathlon.	Decrease in oxidative stress due to 6–13% decrease in proinflammatory cytokines and 55% increase in anti-inflammatory cytokines, after intense exercise. 24–69% increase in plasma amino acids and athletic performance in the probiotic group, due to improved fatigue index (FI) and maximal anaerobic power (PP).	Supplementation with *Lactobacillus plantarum PS128* may be a potential ergogenic aid for better training management, physiological adaptations to exercise and health promotion.
Marshall et al. 2017 [24]	Double-blind RCT in 32 ultramarathoners (6 women and 26 men) High quality = 5 N = 11–50	Probiotic group (*n* = 11): 150 g/day of *Lactobacillus acidophilus*, *bifidobacterium bifidum* and *bifidobacterium aniamles subspecies lactis*. Probiotic + glutamine group (*n* = 10): 0.9 g L-glutamine per 5 g dosage. No supplementation (*n* = 11). Follow-up during the 12 weeks prior to the Sabre Marathon.	After the run, eHsp72 concentrations increased by 124% (F [1,3] = 22.716, *p* < 0.001), indicating increased levels of systemic stress. There was no difference between groups in eHsp72 concentration.	Supplementation with probiotics or probiotics + glutamine did not decrease the concentration of eHsp72, an indicator of systemic stress after an ultramarathon.
Strasser et al. 2016 [25].	Double-blind RCT in 29 ultramarathoners (16 women and 13 men) High quality = 5 N = 11–50	1 sachet before breakfast (1 × 10^10^ CFU of *Bifidobacterium bifidum* *W23,* *Bifidobacterium lactis* *W51,* *Enterococcus faecium* *W54,* *Lactobacillus acidophilus* *W22,* *Lactobacillus brevis* *W63*, and *Lactococcus lactis* *W58*), for 12 weeks.	Tryptophan levels after exercise remained unchanged in the probiotic group. The proportion of placebo who suffered 1 or 2 symptoms of URTIs increased 2.2-fold compared to the probiotic group (0.79 vs. 0.35; *p=* 0.02).	Daily probiotic supplementation decreased the rate of exercise-induced tryptophan degradation and reduced the incidence of URTIs, but did not benefit athletic performance, although training load was higher (h/week).
Jager et al. 2016 [26].	Double-blind RCT in 15 endurance runners (male) High quality = 5 N = 11–50	1 capsule/day of *Bifidobacterium breve BR03* and *Streptococcus thermophilus FP4* (5 × 10^9^ CFU) for 3 weeks.	Probiotic supplementation decreased circulating IL-6 up to 48 h after exercise. It improved the average peak torque in an isometric test at 24 and 72 h.	Probiotic intake mitigates performance reductions and muscle tension in the days after exercise, which damages muscles. Specific dietary probiotics can aid in performance recovery after heavy eccentric exercise, which influences the performance of endurance runners.
Jager et al. 2016 [27]	Double-blind RCT in 20 endurance runners (male) High quality = 5 N = 11–50	Cross-over study: Week 0–2, supplementation with 20 gr of casein after breakfast. Week 4–6, supplementation with *Bacillus coagulans GBI-30*, 6086 + 20 gr casein after breakfast.	Probiotic + casein supplementation increased recovery at 24 and 72 h and decreased pain at 72 h, with an increase in creatine kinase (CK) of +137.7% (*p* = 0.001) versus casein group, which was +266.8% (*p* = 0.0002). In addition, muscle damage decreased (*p* = 0.08). Intense exercise maintained athletic performance (+10.1 watts + 1.7%).	Probiotic + casein supplementation decreased muscle damage levels, increased recovery and maintained performance after damaging exercise.
Gleeson et al. 2016 [28].	Double-blind, placebo-controlled RCT in 243 collegiate athletes (women and men) High quality = 6	2 drinks/day (breakfast and dinner) of *Lactobacillus casei shirota* (6.5 × 10^9^ CFU) for 20 weeks.	There were no significant differences in duration and severity of URTIs between groups. Significant interaction effect between times and groups against CMV^1^⁰ and EBV^1^⁰ antibodies in plasma (*p* < 0.01), in the probiotic group.	Probiotic supplementation did not decrease the incidence of URTIs, but decreased EBV and CMV antibodies.

RCT: randomized clinical trial; CFU: colony-forming units; URTIs: upper respiratory tract infections; VO2max: peak oxygen volume; EBV: Epstein–Barr virus; CMV: cytomegalovirus.

## Data Availability

Data was collected from the available literature. Data for perfoming the systematic review are available in the manuscript’s tables.

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
