# Peer review of "Impact of Probiotics on the Performance of Endurance Athletes: A Systematic Review"

_ijerph, 2021, doi:10.3390/ijerph182111576_

Round 1

Reviewer 1 Report

Thank you for the opportunity to review this manuscript. 

I recommend highlighting the novelty of your  review. 

I recommend extending the conclussion to underline the main findings and ideeas. 

Author Response

Dear reviewer,

following your comments, we have expanded the conclusions and highlighted the novelty of the manuscript, as no systematic reviews of the scientific literature had been carried out to date. We wanted to analyse the existing evidence and not just show the results, before affirming or denying the benefits of the use of probiotics in endurance athletes.

Again, thank you for your comments, which will help to improve our manuscript.

Kind regards.

Reviewer 2 Report

    This article summarized the studies about the effects of probiotics supplements on endurance athletes and tried to make a conclusion whether the effects are positive or not. Although the article in-depth analyzed the correlation between the types, duration, dosage of probiotics and athletes’ health status such as immune system, oxidative stress and GI problems, it lacks quantified analysis such as forest plot and funnel diagram, which were the most important to ensure the quality and reliability of a meta-analysis. In our opinion, this article is a simple systematic review with an incomplete meta-analysis.

1 Line 131: the table array title is not so professional for a meta-analysis. Please refer to some excellent meta-analysis article. What’s important, forest plot and funnel diagram must be included.

2 From this article especially in the Introduction, there is no research conflicts about the effects of probiotics supplements on endurance athletes. Thus, it seems meaningless to conduct a meta-analysis. In addition, it’s better to include the side-effects during the probiotics supplement treatment. Here, the research conflicts refer to existed positive and negative outcomes.

3 The part of Discussion should be divided into several subtitles such as dosage form, health issues and current limitations. In addition, the effects of probiotics on other aspects related to endurance athletes should be discussed in details, which is necessary to increase the theoretical depth and comprehensiveness of the article. Here in this article, only VO2 max, immune response, GI problem, oxidative stress and inflammation were discussed. Are there any other aspects? As we know, probiotics are involved in many pathways including gut metabolites, systematic hormones, nerve system and others.

4 This article pointed out many shortcomings about the current studies, whereas no constructive perspectives were proposed.

Author Response

Estimado revisor,

después de leer sus comentarios, se han agregado algunas modificaciones al manuscrito.

The title of table 2 has been modified, following the parameters of other systematic reviews and analyses. The disparity of the methodology used, as well as the conclusions of the different studies, have limited us from carrying out a meta-analysis, so the authors opted for a systematic review, but with very strict methodological criteria, following the PRISMA recommendations.

In the introduction, a paragraph has been added to justify the need for this systematic review, as the consumption of probiotics has increased considerably at a global and national level (https://elpais.com/economia/2020-12-30/bacterias-que-alimentan-la-creciente-industria-de-los-probioticos.html). Their recommendations are sometimes not evidence-based.

No se agregaron otros conceptos, ya que esta revisión sistemática solo discutió los resultados obtenidos, por lo que la exhaustividad está en línea con estos resultados. Además, nos resultó muy difícil distinguir entre párrafos.

En 2016 se publicó una revisión (https://jissn.biomedcentral.com/articles/10.1186/s12970-016-0155-6), que, aunque aparece en el título como una revisión sistemática, parece más una revisión narrativa, porque no refleja la metodología sistemática analizada y por lo tanto su discusión se dividió en secciones debido a su amplitud. Tomamos sus comentarios como una ayuda para el desarrollo de una revisión narrativa o documentación didáctica en este campo.

La sección de conclusiones se ha ampliado y se hacen sugerencias de una manera más constructiva para futuras investigaciones en este campo.

Gracias nuevamente por sus comentarios, que ayudan a mejorar el manuscrito.

Atentamente.

Reviewer 3 Report

Just very few and not substantial observations 

  • Line 62: better write: “some” instead “many.. elite athletes” We don’t have data about the number of athletes that do this choice
  • Line 72-73: There is only one paper about the increase of Veillonella and only after marathon, so it is better to write that exercise in general, may cause this adaptation.
  • Discussion: it could be more emphasize the very likely interrelation between nutrition, hydration and pre and pro-biotics. It is not a necessary insertion, the paper is already exhaustive, but in a so complete dissertation, the absence of certainty should be remarked.

Author Response

Dear reviewer,

following your comments, modifications have been made to the manuscript.

We appreciate these comments, as they help to improve the quality of the manuscript and the understanding of future readers, should it eventually be published.

Kind regards.

Reviewer 4 Report

Dear authors,

As a researcher, I am annoyed by the recent multiplication of systematic reviews, the advantages of which are almost inexistant when they survey less than two dozens of articles. Conclusions drawn from such a restricted panel can only be utterly speculative and could be used to pave the way for a thorough study on a given subject, but not for a full-blown article.

Here, we're confronted with a text that I admit is well-written, detailed and full of solid references, but which relevance has yet to be proven. There are so much differences between the nine studies retained that nothing can be genuinely inferred. The probiotics supplemented are never the same, neither are the doses, the supplementation procedures, the population under study or the sporting disciplines, and the list goes on.

I'm not fundamentally opposed to the publication of the present article but I really wonder what benefits it can bring to the scientific community, except the proof that the field of research "Probiotics and Sport" is a gigantic mess.

Aside from this, I noted two typing/language mistakes:

L. 98: double WW (WWe)

l. 265-268: "A study carried out on university athletes supplemented with probiotics (Lactobacillus gasseri OLL2809), which eluded the decrease in the action of NK cells after strenuous exercise, resulting in an improvement in defences against infections and elevation of mood [41]."

This sentence is not very well written and we don't know what the "which" refers to (probiotics OLL2809 or the study?).

Author Response

Dear reviewer.

We are grateful for your comments and would like to express the reason for this review.

In recent years, the consumption of probiotics has increased considerably. In Spain, the consumption of commercial probiotic preparations has increased from 73 million in 2016 to 121 million in 2019 (https://elpais.com/economia/2020-12-30/bacterias-que-alimentan-la-creciente-industria-de-los-probioticos.html). Many professionals recommend their use for health problems, especially those related to the gastrointestinal tract. Following a search in Pubmed, using the Mesh "Probiotics", the number of publications from 1947 to 2015 was 16,145, increasing considerably in the last 6 years (18,498 new studies). This data shows us the boom of studies in this field.

This fact, together with the search for supplements that can improve sports performance in endurance athletes, makes it a supplement valued by many athletes and coaches.

The aim of this review is to comprehensively compile all the evidence related to probiotics and sports performance, and to serve as a support for colleagues who wish to carry out studies in this field. In addition, as he comments, it can serve as a starting point for conducting exhaustive Controlled Clinical Trials that correct the biases present and can provide new evidence. A review was published in 2016 (https://jissn.biomedcentral.com/articles/10.1186/s12970-016-0155-6), which, although it appears in the title as a systematic review, seems more like a narrative review, because it does not reflect the systematic methodology analysed and hence its discussion was divided into sections due to its breadth.

We understand that you may have your doubts due to the number of articles present in the review, but they are a reflection of the existing publications. This number may be due to the lack of studies in this field, the lack of scientific rigour in some studies (only studies of high scientific quality were selected), and the quality criteria selected by the authors.

The disparity of strains, disciplines, ..., helps readers in this field to understand the difficulties in reaching a valid conclusion to put into practice.

This field of research is not a disaster, but it is an under-studied and under-evidenced field. Perhaps this is the greatest benefit of this review, the compilation of the highest quality studies, and the conclusion that there is no strong evidence for the use of probiotics in sports performance enhancement. This conclusion is a starting point for athlete counselling and future research.

In addition, the typos have been corrected and we apologise, as we should have been vigilant in ensuring that this did not happen.

Thank you again for your comments, and we hope we have resolved your concerns.

Kidn regards.

Round 2

Reviewer 2 Report

1 A complete meta-analysis could not be carried out according to the concerns presented by the authors, so it is all right not to include forest plot and funnel diagram. However, the side-effects or negative results should be included when endurance athletes are consuming probiotics. In addition, as the author introduced, the studies on this topic are relatively few, so this review needs to systematically list what have known and what need to be known when endurance athletes are consuming probiotics. The structure of this article is not clear and is hard to understand.

2 We could not see the replies to our questions proposed at the first time:

(1) In this article, only VO2 max, immune response, GI problem, oxidative stress and inflammation were discussed. Are there any other aspects? As we know, probiotics are involved in many pathways including gut metabolites, systematic hormones, nerve system and others.

(2) The part of Discussion should be divided into several subtitles such as dosage form, health issues and current limitations.

Author Response

Dear reviewer

thank you again for your comments which help us to improve the manuscript. In the following, we respond to your comments and add the suggested changes to the manuscript.

1 A complete meta-analysis could not be carried out according to the concerns presented by the authors, so it is all right not to include forest plot and funnel diagram. However, the side-effects or negative results should be included when endurance athletes are consuming probiotics.

No side effects associated with the use of proiotics were found in the studies reviewed. A related paragraph has been added.

In addition, as the author introduced, the studies on this topic are relatively few, so this review needs to systematically list what have known and what need to be known when endurance athletes are consuming probiotics.

A paragraph has been added on this point, linking it to the previous one.

The structure of this article is not clear and is hard to understand.

The structure of the article has been modified, especially in the discussion section.

2 We could not see the replies to our questions proposed at the first time:

(1) In this article, only VO2 max, immune response, GI problem, oxidative stress and inflammation were discussed. Are there any other aspects? As we know, probiotics are involved in many pathways including gut metabolites, systematic hormones, nerve system and others.

As you comment, we have not delved into other effects of probiotics because of the scope of this review, avoiding it being a review of probiotics alone or naming effects in endurance athletes that have not been evidenced in previous studies. A point on mental health has been added, as this would be very interesting and could influence the performance of athletes.

(2) The part of Discussion should be divided into several subtitles such as dosage form, health issues and current limitations.

The discussion section has been divided to make it more attractive for the future reader.

Thank you for your comments and time spent on our manuscript.

Kind regards

Reviewer 4 Report

Except for a few sentences that actually sound like an acknowledgement that this meta-analysis bears very few significance given the current landscape in that particular field of research, I don't fathom why this version would be more relevant than the former.

However, considering that the three other reviewers gave positive/medium assessments, and out of respect for them and the amount of time spent by the authors, I will give my green light. Pale green, but still.

Author Response

Dear reviewer,
we appreciate your comments. Further modifications suggested by another reviewer have been made, hopefully bringing us closer to your expectations. 
We believe that this review not only addresses the lack of evidence but should be a starting point for those researchers who, after reading it, may initiate a research project in this field.
Kind regards